# Sarcopenia and Functional Decline in Postmenopausal Women: The Roles of Type 2 Diabetes and Physical Activity

**DOI:** 10.3390/medsci13040268

**Published:** 2025-11-14

**Authors:** Anthony Rodrigues de Vasconcelos, Fernando José de Sá Pereira Guimarães, Pedro Weldes da Silva Cruz, Maria Joana Mesquita Cruz Barbosa de Carvalho, Aline de Freitas Brito, Keyla Brandão Costa, Lucas Savassi Figueiredo, Paulo Adriano Schwingel, Denise Maria Martins Vancea, Manoel da Cunha Costa

**Affiliations:** 1Graduate Program in Rehabilitation and Functional Performance (PPGRDF), University of Pernambuco, Petrolina 56328-900, PE, Brazil; fernando.guimaraes@fmo.edu.br (F.J.d.S.P.G.); aline.brito@upe.br (A.d.F.B.); manoel.costa@upe.br (M.d.C.C.); 2Research Group on Physical Exercise and Non-Communicable Chronic Diseases (GPEFDCNT), University of Pernambuco, Recife 50100-130, PE, Brazil; pedro.weldes@upe.br (P.W.d.S.C.); keyla.costa@upe.br (K.B.C.); savassi88@hotmail.com (L.S.F.); denise.martins@upe.br (D.M.M.V.); 3High School Physical Education (ESEF), University of Pernambuco, Recife 50100-130, PE, Brazil; 4School of Sports (FADEUP), University of Porto, 4200-450 Porto, Portugal; mjoanacarvalho@reit.up.pt; 5School of Physical Education, Physiotherapy and Occupational Therapy (EEFFTO), Federal University of Minas Gerais (UFMG), Belo Horizonte 31270-901, MG, Brazil; 6Laboratório da Avaliação da Performance Humana (LapH), University of Pernambuco, Recife 50100-130, PE, Brazil

**Keywords:** sarcopenia, diabetes mellitus, type 2, exercise, postmenopause, walking speed

## Abstract

Background/Objectives: Postmenopausal women face an elevated risk of sarcopenia and functional decline, yet the distinct roles of type 2 diabetes mellitus (T2DM) and physical inactivity in these outcomes remain unclear. This study aimed to investigate the independent and combined associations of T2DM and physical activity on sarcopenia and functional performance in postmenopausal women. Methods: This was a cross-sectional study of 175 postmenopausal women stratified by T2DM status and physical activity level (active ≥150 min/week vs. insufficiently active). Body composition was assessed via dual-energy X-ray absorptiometry, muscle strength by handgrip dynamometry, and functional performance by gait speed. Sarcopenia was diagnosed using the Asian Working Group for Sarcopenia 2019 criteria. Binary logistic regression calculated odds ratios (ORs) for adverse outcomes. Results: Physical inactivity was the strongest predictor of functional decline, with insufficiently active women showing nearly four-fold increased odds of slow gait speed (<1.0 m/s) compared to active counterparts (OR: 3.93; 95% CI: 1.24–12.45). While T2DM appeared protective against sarcopenia in unadjusted analysis, multivariate adjustment revealed obesity (OR: 4.97; 95% CI: 1.62–15.20) and T2DM (OR: 3.80; 95% CI: 1.59–9.08) as independent sarcopenia predictors. Conclusions: Distinct associational profiles emerged for sarcopenia and functional decline in postmenopausal women. While T2DM and obesity are independently associated with sarcopenia through metabolic mechanisms, physical inactivity emerged as the strongest predictor of functional impairment. These findings support targeted interventions: metabolic optimization for muscle mass preservation and structured physical activity, particularly resistance training, for maintaining functional independence in this high-risk population.

## 1. Introduction

The menopausal transition is characterized by a substantial decline in estrogen, which promotes an unfavorable redistribution of body composition, including an increase in visceral fat and a decrease in muscle mass [1]. These hormonal shifts contribute to a pro-inflammatory state and insulin resistance, significantly heightening the susceptibility of postmenopausal women to cardiometabolic diseases such as type 2 diabetes mellitus (T2DM) [2,3]. T2DM, a condition affecting over 530 million people globally, is notably influenced by aging and endocrine changes, creating a high-risk profile in this population [4].

A critical consequence of the interplay between aging, hormonal changes, and metabolic disease is the acceleration of sarcopenia [5,6,7]. Sarcopenia is particularly prevalent and severe in individuals with T2DM, where studies show muscle loss is more pronounced compared to normoglycemic peers [8,9]. Persistent hyperglycemia and associated metabolic alterations, such as the accumulation of advanced glycation end-products (AGEs), exacerbate muscle deterioration [10]. This decline in muscle health impairs the ability to perform activities of daily living (ADLs) and increases the risk of disability, falls, and functional dependence [5,6,7,11].

Conversely, adopting an active lifestyle, particularly through physical training, is a primary strategy to mitigate these risks. Exercise has been shown to improve glycemic control and contribute to maintaining appendicular muscle mass, thereby enhancing muscle strength and overall physical fitness [12]. This helps prevent the detrimental effects of sarcopenia and its functional impairments [12,13,14,15].

While recent research highlights the contribution of other behavioral factors, such as emotional eating, to T2DM risk in younger adult populations [16], physical inactivity remains the predominant modifiable risk factor in older, postmenopausal women [17,18]. However, while the negative impacts of T2DM [8,9,10] and physical inactivity [17,18] are known individually, their independent and combined associations with musculoskeletal health in this specific high-risk group are less clear. It is particularly unknown whether these two factors (one metabolic, one behavioral) contribute equally to muscle loss (sarcopenia) versus functional decline (e.g., slow gait speed), or if they have distinct associational profiles. Furthermore, the relationship between T2DM and sarcopenia is often confounded by the high prevalence of obesity in this population [19], potentially masking the true associations.

Therefore, the present study had two primary aims. First, using a stratified cross-sectional design, we aimed to analyze the independent associations of T2DM and physical inactivity with sarcopenia and functional performance. Second, we sought to investigate the confounding role of obesity in the T2DM–sarcopenia relationship [19], testing the hypothesis that these metabolic and behavioral factors would emerge as distinct predictors for different musculoskeletal outcomes.

## 2. Materials and Methods

### 2.1. Study Design and Ethical Approval

This cross-sectional study was designed and conducted in accordance with the translated version of the Strengthening the Reporting of Observational Studies in Epidemiology (STROBE) guidelines [18]. This research received ethical approval from the Research Ethics Committee at the University of Pernambuco—UPE (Report Number. 2.332.880; CAAE: 72113417.2.0000.5192). All procedures were conducted in accordance with the ethical standards outlined in the Declaration of Helsinki (1964, revised in 2013) and complied with the Brazilian National Health Council Resolutions 466/2012 and 510/2016. Written informed consent was obtained from all individuals before their inclusion in the study.

### 2.2. Participants and Recruitment

A total of 175 postmenopausal women were recruited for this study (Figure 1). Participants were sourced through public advertisements on social media and posters, direct phone invitations facilitated by the UPE, and from an existing cohort in the “Doce Vida” Supervised Physical Exercise Program for Diabetics, an extension program of the university [13].

A sample size calculation was performed a priori using OpenEpi software (release 3.03, 2014) [20,21] to ensure sufficient power for our primary outcomes. The calculation was based on detecting a clinically significant difference in the prevalence of sarcopenia between key groups. Based on preliminary data and related literature, we anticipated a prevalence of approximately 30% in the inactive group and 10% in the active group. To achieve a statistical power of 80% (1-β) with a 95% confidence level (α = 0.05), a minimum of 165 participants was required. The final enrolled sample of 175 participants exceeded this a priori requirement, providing sufficient power for the primary analyses.

Participants were required to meet the following criteria: (a) women aged 40 years and older, (b) residing in Recife or its metropolitan area, (c) who appeared to be healthy, (d) had no limitations for motor tests, (e) self-reported postmenopausal status, (f) not undergoing cancer treatment, and (g) did not use medications known to alter body composition or prosthetics. Participants were considered ineligible if they: (a) failed to complete the Dual-Energy X-ray Absorptiometry (DXA) scan, (b) failed to complete any of the required functional performance tests, or (c) tested positive for Human Immunodeficiency Virus (HIV).

### 2.3. Group Stratification

Participants were stratified into four distinct groups based on their T2DM diagnosis and physical activity level:Group 1 (G1): Physically active women with T2DM. These participants were enrolled in the “Doce Vida” program, engaging in supervised combined (resistance and aerobic) training sessions three times per week, with each session lasting approximately 90 min.Group 2 (G2): Insufficiently active women with T2DM who did not engage in regular, structured physical activity.Group 3 (G3): Physically active normoglycemic women. Their activity consisted of unsupervised walking (average 20 min/session) combined with a supervised water-based exercise program (approximately 50 min/session), performed three times per week.Group 4 (G4): Insufficiently active normoglycemic women who did not participate in regular physical activity.

### 2.4. Procedures and Data Collection

Data collection was conducted in two phases at the Human Performance Assessment Laboratory (LapH) and Biomechanics Laboratory (LABi) from the UPE. In the first phase, participants were screened for eligibility and provided with instructions. In the second phase, a structured questionnaire was administered to collect sociodemographic data, medical history, T2DM diagnosis and duration, and physical activity details. Subsequently, a series of anthropometric and functional assessments were performed.

### 2.5. Outcome Measures and Definitions

The primary exposure variables were T2DM status, confirmed according to the Brazilian Diabetes Society guidelines [4], and physical activity level, dichotomized as physically active (≥150 min/week of moderate-intensity exercise) or insufficiently active (<150 min/week). The primary outcomes were sarcopenia and slow gait speed [22].

#### 2.5.1. Anthropometric and Body Composition

Participants’ total body mass was measured in kilograms (kg), and height was measured in centimeters (cm) using a properly calibrated anthropometric scale (PL-200, Filizola S.A., São Paulo, SP, Brazil), adhering to the standards set by NBR ISO/IEC 17025:2005. Body mass index (BMI) was calculated by dividing body mass (kg) by the square of height (m^2^).

Body composition was assessed using a DXA system Hologic Discovery Wi Bone Densitometry (Hologic, Inc., Marlborough, MA, USA). The following indices were calculated:(a)Appendicular Skeletal Muscle Index (ASMI): Calculated by dividing appendicular lean mass (kg) by the square height (m^2^). Low muscle mass was defined as an ASMI < 5.4 kg/m^2^.(b)Fat Mass Index (FMI): Calculated by dividing total fat mass (kg) by the square of height (m^2^). Obesity was defined as an FMI ≥ 13.0 kg/m^2^.(c)Fat Mass Percentage (FM%).

#### 2.5.2. Functional Performance

Gait speed was assessed during a single trial of usual walking speed over a 5 m walkway with designated acceleration and deceleration zones. Slow gait speed was defined as <1.0 m/s [22].

Grip strength was measured using a Jamar hydraulic hand dynamometer model 5030 J1 (Sammons Preston Rolyan, Bolingbrook, IL, USA). Participants were tested while standing, and the maximum value from one measurement on each hand was recorded. Low muscle strength was defined as <18 kgf [22].

#### 2.5.3. Sarcopenia Diagnosis

Sarcopenia was diagnosed according to the Asian Working Group for Sarcopenia (AWGS) 2019 consensus criteria [22]. The AWGS 2019 framework was selected over other international guidelines (e.g., EWGSOP2 [23]) because its functional thresholds (handgrip strength < 18 kg; gait speed < 1.0 m/s) are more conservative than those of EWGSOP2 (handgrip strength < 16 kg; gait speed ≤ 0.8 m/s) [23], providing enhanced sensitivity for detecting early functional decline—a critical consideration for this study’s focus on identifying incipient sarcopenia in a metabolically vulnerable postmenopausal population with and without T2DM.

According to AWGS 2019 criteria [22], sarcopenia was defined as the presence of low ASMI (<5.4 kg/m^2^) combined with either low muscle strength (handgrip strength < 18 kg) or slow gait speed (<1.0 m/s). Participants meeting these criteria were classified as having sarcopenia; those not meeting the criteria were classified as non-sarcopenic.

### 2.6. Statistical Analysis

Data were double-entered and analyzed using the SPSS Statistics for Windows (IBM Corp., Armonk, NY, USA, release 22.0, 2013). The normality of continuous variables was assessed using the Kolmogorov–Smirnov test, while Levene’s test was employed to examine the homogeneity of variances. Continuous variables were summarized with means and standard deviations (SDs), while categorical variables were presented as absolute (n) and relative (%) frequencies. Baseline characteristics between the four investigated groups were compared using one-way Analysis of Variance (ANOVA). Pearson’s chi-square test (χ^2^) or Fisher’s exact test was employed as appropriate for categorical variables. Binary logistic regression analyses were conducted to calculate odds ratios (ORs) and 95% confidence intervals (CIs) to evaluate the association between risk factors (T2DM, physical inactivity, obesity) and dichotomous outcomes (sarcopenia, slow gait speed). All *p*-values and 95% CIs were calculated and reported with exact values. A two-tailed significance level of 5% (*p* ≤ 0.05) was adopted for all statistical tests.

## 3. Results

### 3.1. Baseline Characteristics of the Study Sample

The final study sample comprised 175 postmenopausal women, of whom 74 (42.3%) had a diagnosis of T2DM. The sample was stratified into four groups based on T2DM status and physical activity level. Among postmenopausal women with T2DM, a majority (60.8%) were classified as physically active, whereas among normoglycemic women, the majority (58.4%) were classified as insufficiently active.

The baseline anthropometric, body composition, and physical performance characteristics of the four groups are detailed in Table 1. Significant between-group differences were observed for the ASMI (*p* = 0.002), handgrip strength (*p* = 0.008), and gait speed (*p* = 0.002).

The prevalence of adverse clinical outcomes is presented in Table 2. Notably, there were significant differences in the prevalence of slow gait speed (*p* < 0.001) and sarcopenia diagnosis (*p* = 0.008) across the groups.

### 3.2. The Confounding Role of Obesity in the Association Between T2DM and Low Muscle Mass

In the unadjusted analysis, a paradoxical association was observed between T2DM and muscle mass. Women with T2DM had a significantly lower prevalence of low muscle mass compared to their normoglycemic counterparts (17.6% vs. 32.7%; *p* = 0.03), exhibiting 56% lower odds of this outcome (OR: 0.44; 95% CI: 0.21–0.91), as shown in Table 3.

However, to investigate this relationship further, a binary logistic regression model was performed, adjusting for glycemic status, obesity (defined by FMI), and physical activity level. In this adjusted model, the protective association of T2DM was no longer present. Instead, obesity emerged as the strongest predictor, significantly increasing the odds of low muscle mass by more than five-fold (OR: 5.48; 95% CI: 1.82–16.47; *p* = 0.002). T2DM showed a non-significant trend towards increased odds (OR: 2.05; 95% CI: 0.94–4.46; *p* = 0.07), while physical activity level was not significantly associated with low muscle mass (OR: 0.74; 95% CI: 0.36–1.52; *p* = 0.41). These results suggest that obesity acts as a critical confounding variable, masking the underlying detrimental relationship between T2DM and muscle mass.

### 3.3. Physical Inactivity as a Primary Driver of Functional Decline

Physical activity level was strongly associated with functional performance. As detailed in Table 4, postmenopausal women who were insufficiently active had nearly four times greater odds of exhibiting clinically slow gait speed (<1.0 m/s) compared to their physically active peers (OR: 3.93; 95% CI: 1.24–12.45; *p* = 0.01).

### 3.4. Independent Predictors of Sarcopenia

Similar to the findings for low muscle mass, the unadjusted analysis revealed that sarcopenia was significantly less prevalent among women with T2DM compared to normoglycemic women (10.8% vs. 31.7%; *p* < 0.001). This corresponded to 74% lower odds of sarcopenia in the T2DM group (OR: 0.26; 95% CI: 0.11–0.61), as shown in Table 5.

To elucidate the independent predictors of sarcopenia, a binary logistic regression was conducted, adjusting for glycemic status, obesity, and physical activity. In the adjusted model, the relationship was reversed. Both T2DM (OR: 3.80; 95% CI: 1.59–9.08; *p* = 0.003) and obesity (OR: 4.97; 95% CI: 1.62–15.20; *p* = 0.005) were identified as significant and independent predictors of increased odds of sarcopenia. Physical inactivity was not significantly associated with sarcopenia in this model (OR: 0.67; 95% CI: 0.31–1.45; *p* = 0.31). These findings confirm that both diabetes and obesity independently contribute to the risk of sarcopenia, a relationship that was obscured in the unadjusted analysis.

## 4. Discussion

This cross-sectional study reveals a fundamental dichotomy in the pathophysiology of musculoskeletal decline in postmenopausal women: while T2DM and obesity are independently associated with development of sarcopenia through metabolic pathways, physical inactivity emerges as the predominant predictor of functional impairment. This distinction has profound implications for clinical practice, suggesting that preserving muscle mass and maintaining functional capacity may require distinct therapeutic strategies.

### 4.1. The Sarcopenia Paradox: Associational Profiles of Obesity and Exercise Modality

Our most striking finding was the apparent protective association of T2DM with low muscle mass in unadjusted analyses—a paradox that resolved only after controlling for body composition. This highlights a critical issue in sarcopenia research: the statistical masking effect of obesity, a condition highly prevalent in T2DM populations [19]. After adjusting for FMI, the protective association disappeared, and T2DM emerged as an independent predictor of sarcopenia (Table 5) [8,9,10,24].

This “protective” paradox is likely twofold. Beyond the statistical confounding by obesity [19], it may reflect a complex physiological mechanism: the “mass-function decoupling” characteristic of dynapenia [10]. In the chronic hyperinsulinemic state of T2DM, insulin’s potent anti-catabolic role (i.e., the inhibition of muscle protein breakdown) may contribute to the preservation of muscle mass [8,9]. However, this tissue is often dysfunctional, insulin-resistant, and infiltrated by fat (myosteatosis) [8,9]. Our unadjusted analysis likely captured this preserved-but-dysfunctional mass, whereas the adjusted model revealed the underlying detrimental association.

The mechanistic basis for this detrimental association is well-supported, involving the accumulation of advanced glycation end-products (AGEs) in muscle tissue, impaired contractile function, insulin resistance, mitochondrial dysfunction, and chronic inflammation [8,9,10].

Furthermore, this heterogeneity in exercise modality, initially noted as a potential limitation, reveals a critical finding. The physically active T2DM group (G1) engaged in supervised combined training (resistance and aerobic), whereas the normoglycemic active group (G3) performed predominantly aerobic and water-based activities. Notably, the G1 exhibited superior appendicular skeletal muscle mass (ASMI: 6.5 ± 0.8 kg/m^2^) compared to their metabolically healthy G3 counterparts (ASMI: 5.7 ± 0.8 kg/m^2^) (Table 1). This finding suggests that the anabolic stimulus from resistance training in G1 was sufficiently potent not only to mitigate the catabolic effects of T2DM [25] but to produce superior muscle mass outcomes compared to the healthy group performing aerobic-only exercise [14]. This strongly supports our conclusion advocating for precision in exercise prescription and aligns with meta-analyses and randomized controlled trials demonstrating that combined training is superior to aerobic-only modalities for improving body composition in postmenopausal women [14,25].

### 4.2. Physical Inactivity: The Primary Predictor of Functional Decline

While the relationships between T2DM, obesity, and sarcopenia were complex and required multivariate adjustment to elucidate, the impact of physical inactivity on functional performance was unequivocal. Insufficiently active women demonstrated nearly four-fold higher odds of clinically slow gait speed (<1.0 m/s), a threshold associated with increased mortality, hospitalization, and loss of independence [10,17,18,26,27].

The strength of this association underscores a critical concept: functional capacity reflects not merely muscle mass but the integrated performance of neuromuscular, cardiovascular, and metabolic systems [17,18]. Our findings align with Sardinha et al. [17], who demonstrated that fitness parameters predict physical independence more accurately than body composition measures alone. This suggests that while sarcopenia represents structural deterioration at the tissue level, gait speed captures the functional consequences of systemic deconditioning.

The clinical significance of reduced gait speed extends beyond mobility. Recent evidence links slow gait speed with cognitive decline [27], increased depression risk [18], and higher T2DM incidence [28], creating a vicious cycle of functional and metabolic deterioration. Our data support the growing consensus that gait speed should be considered the “sixth vital sign” in geriatric assessment [18,29,30,31].

### 4.3. Integrating Metabolic and Neuromuscular Pathways

The dissociation between sarcopenia risk factors (T2DM and obesity) and functional decline predictors (physical inactivity) revealed in our study challenges the traditional conflation of these outcomes. While previous research has often treated sarcopenia and functional impairment as parallel consequences of aging and disease [5,6,7], our findings suggest they may represent distinct phenotypes governed by separate, though potentially interactive, pathways.

The strong, independent associations of T2DM and obesity with sarcopenia in our adjusted models point to a metabolic etiology. This pathway is characterized by systemic factors known to compromise muscle quality and quantity. Chronic hyperglycemia and insulin resistance trigger mitochondrial dysfunction and promote the accumulation of Advanced Glycation End-Products (AGEs) within muscle tissue [10]. Concurrently, chronic low-grade inflammation (inflammaging) impairs myogenesis and protein synthesis while promoting ectopic lipid infiltration (myosteatosis) [8,9]. These processes lead to progressive structural and metabolic deterioration of skeletal muscle, independent of physical activity levels. This aligns with evidence from Hu et al. [5], who demonstrated that sarcopenia predicts cognitive impairment independently of physical activity, and Li et al. [6], who found persistent associations between sarcopenia and depression after controlling for functional status.

In contrast, the exclusive association of physical inactivity with slow gait speed in our study suggests a predominantly neuromuscular mechanism. Gait speed is a complex motor task that relies not solely on muscle mass but on efficient integration of the entire neuromuscular system, including optimized motor unit recruitment, intact neuromuscular junction integrity, and precise intermuscular coordination [27,29,30]. Physical disuse impairs these neural activation patterns through deconditioning, leading to compromised functional performance even when muscle mass is preserved. Thus, while women with T2DM may experience metabolic muscle deterioration, physically inactive women demonstrate impaired neuromuscular performance, supporting evidence that cardiorespiratory fitness and neuromuscular efficiency, rather than mass alone, predict functional independence [17,18].

These distinct yet potentially interactive pathways have important clinical implications, suggesting that comprehensive management of postmenopausal women with T2DM requires dual targeting: metabolic optimization to preserve muscle tissue integrity and structured physical activity to maintain neuromuscular function and functional capacity.

### 4.4. Clinical and Public Health Implications

Our findings necessitate a paradigm shift in managing postmenopausal women with T2DM. Current clinical guidelines focus predominantly on glycemic control, with insufficient attention to functional outcomes that directly impact quality of life and independence [32,33]. The dissociation between metabolic and functional risk factors identified in our study calls for comprehensive care models that address both dimensions simultaneously.

In clinical practice, the integration of functional assessments into routine diabetes care emerges as an immediate priority. Gait speed measurement, requiring less than two minutes to perform, provides a powerful screening tool for disability risk that complements traditional metabolic monitoring [10,18,34]. This simple assessment could identify women at the highest risk for functional decline before irreversible disability occurs. Furthermore, our findings emphasize the need for precision in exercise prescription. Rather than generic recommendations to “be more active,” clinicians should specify resistance training components for sarcopenia prevention, as supported by recent meta-analyses demonstrating superior outcomes with combined training modalities [25,35]. The identification of sarcopenic obesity in our postmenopausal women groups also highlights the inadequacy of BMI as a sole anthropometric measure, supporting the implementation of body composition assessment through DXA or bioimpedance analysis in high-risk populations [19].

From a public health perspective, these findings argue for fundamental revisions to diabetes management guidelines. The incorporation of functional targets alongside glycemic goals would acknowledge the equal importance of maintaining independence in this population. The development of accessible, supervised exercise programs that include resistance training components represents a critical infrastructure need, particularly given that our physically active T2DM participants who engaged in structured combined training showed better muscle mass preservation than their aerobically active normoglycemic counterparts. Early screening initiatives for sarcopenia and functional decline in postmenopausal women with metabolic disease could enable timely interventions before the onset of disability. Recent evidence from successful community-based programs demonstrates the feasibility and cost-effectiveness of such approaches [36,37], suggesting that implementation barriers are surmountable with appropriate resource allocation and policy support.

### 4.5. Strengths, Limitations, and Future Directions

The present study has several notable strengths, including objective body composition assessment via DXA, systematic group stratification based on both metabolic and behavioral factors, and comprehensive functional evaluation. Importantly, the identification of obesity as a critical confounding variable that masks the T2DM–sarcopenia relationship represents a significant methodological contribution to the field.

However, several limitations warrant consideration. First, the cross-sectional design precludes causal inference; we can only report associations rather than determine etiological pathways. Longitudinal studies are therefore needed to establish temporal relationships between T2DM, physical activity, and musculoskeletal outcomes. Additionally, the heterogeneity in exercise modalities among active groups, while informative, limits direct comparisons and necessitates randomized controlled trials comparing specific training types.

Second, potential selection bias exists due to heterogeneity in recruitment methods. Participants in the active T2DM group (G1) were recruited from a university-affiliated exercise program (“Doce Vida”), whereas participants in the other groups were recruited through public advertisements and social media. The “Doce Vida” cohort may represent a healthy volunteer effect, potentially comprising individuals with greater health literacy, better treatment adherence, and higher intrinsic motivation compared to the broader diabetic population. This limitation restricts the generalizability of our findings to community-dwelling adults with T2DM, and future research should employ more representative community-based sampling strategies.

Third, physical activity was assessed via self-report and dichotomized at the ≥150 min/week threshold. This method is susceptible to recall and social desirability biases, leading to overestimation of actual activity levels compared to objective measures like accelerometry. This may have resulted in misclassification of some participants, potentially diluting the true magnitude of associations between physical activity and functional outcomes. Future research should prioritize objective measurements to confirm these findings.

Fourth, some of our logistic regression models (Table 3, Table 4 and Table 5) produced wide 95% confidence intervals. This indicates that while the associations were statistically significant, the sample size within specific subgroups restricts the precision of these estimates. The true magnitude of the effect (OR) is uncertain and should be interpreted with caution.

Finally, our analysis did not include several key physiological confounders. We did not assess dietary factors (especially protein), vitamin D status, or inflammatory markers, which may modulate the relationships observed [10,37,38,39]. Future studies should incorporate these variables to develop more comprehensive predictive models.

## 5. Conclusions

This cross-sectional study reveals distinct associational profiles for sarcopenia and functional decline in postmenopausal women. Our findings suggest that T2DM and obesity are independently associated with sarcopenia, a relationship obscured by confounding in unadjusted analyses. In contrast, physical inactivity emerged as the strongest independent predictor of functional impairment, specifically slow gait speed.

These findings support a dual-target approach for this high-risk population: (1) metabolic monitoring and management (addressing T2DM and obesity) for muscle mass preservation, and (2) structured physical activity incorporating resistance training (as evidenced by our G1 findings) for maintaining functional independence.

For the growing population of postmenopausal women with T2DM, our results suggest that comprehensive care models should extend beyond glycemic control to encompass functional preservation. Although this cross-sectional design precludes causal inference, our findings provide a strong rationale for longitudinal studies and clinical trials examining whether targeted interventions can prevent disability and preserve quality of life in this vulnerable population.

## Figures and Tables

**Figure 1 medsci-13-00268-f001:**
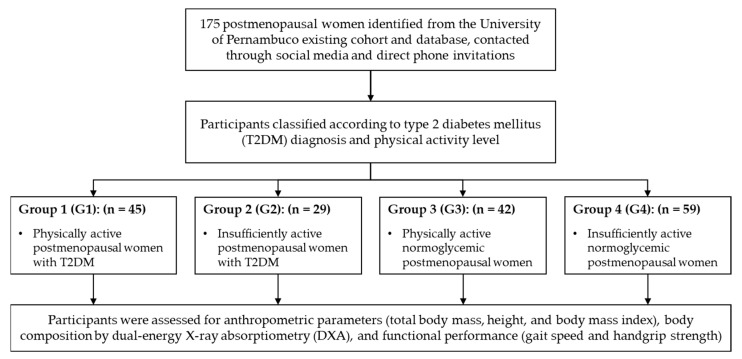
Flow diagram.

**Table 1 medsci-13-00268-t001:** Baseline characteristics of postmenopausal women by diabetes status and physical activity level (*n* = 175).

Variables	G1 (*n* = 45) Mean ± SD	G2 (*n* = 29) Mean ± SD	G3 (*n* = 42) Mean ± SD	G4 (*n* = 59) Mean ± SD	*p*
Age, years	64.5 ± 9.5	65.1 ± 9.2	61.3 ± 10.4	62.1 ± 8.1	0.200
Body mass index, kg/m^2^	27.6 ± 4.2	28.7 ± 4.4	26.7 ± 4.1	26.5 ± 5.1	0.170
Appendicular skeletal muscle index, kg/m^2^	6.5 ± 0.8	6.1 ± 0.8	5.7 ± 0.8	5.7 ± 0.8	0.002
Fat mass index, kg/m^2^	11.0 ± 3.0	11.9 ± 2.8	11.2 ± 4.9	11.4 ± 3.3	0.670
Fat mass, %	39.6 ± 5.7	41.6 ± 3.6	41.6 ± 7.2	42.2 ± 7.3	0.200
Handgrip strength, kgf	21.9 ± 3.7	22.9 ± 5.1	23.8 ± 4.2	21.1 ± 3.6	0.008
Gait speed, m/s	1.4 ± 0.3	1.1 ± 0.3	1.4 ± 0.3	1.4 ± 0.4	0.002

Abbreviations: G1, physically active women with T2DM; G2, insufficiently active women with T2DM; G3, physically active normoglycemic women; G4, insufficiently active normoglycemic women; SD, standard deviation; T2DM, Type 2 Diabetes Mellitus.

**Table 2 medsci-13-00268-t002:** Prevalence of sarcopenia, obesity, and functional impairment by diabetes status and physical activity level (*n* = 175).

Variables	G1 (*n* = 45) *n* (%)	G2 (*n* = 29) *n* (%)	G3 (*n* = 42) *n* (%)	G4 (*n* = 59) *n* (%)	*p*
Obesity (FMI > 13.0 kg/m^2^)	13 (28.9%)	8 (27.8%)	10 (23.8%)	18 (30.5%)	0.900
Low muscle mass (ASMI < 5.4 kg/m^2^)	7 (15.6%)	6 (20.7%)	14 (33.3%)	19 (32.2%)	0.150
Low handgrip strength (<18 kgf)	15 (33.3%)	6 (20.7%)	5 (11.9%)	18 (30.5%)	0.080
Slow gait speed (<1.0 m/s)	0 (0.0%)	10 (34.5%)	4 (9.5%)	4 (6.8%)	<0.001
Sarcopenia diagnosis [22]	5 (11.1%)	3 (10.3%)	11 (26.2%)	21 (35.6%)	0.008

Abbreviations: G1, physically active women with T2DM; G2, insufficiently active women with T2DM; G3, physically active normoglycemic women; G4, insufficiently active normoglycemic women; ASMI, Appendicular Skeletal Muscle Index; FMI, Fat Mass Index; T2DM, Type 2 Diabetes Mellitus.

**Table 3 medsci-13-00268-t003:** Risk of low muscle mass associated with type 2 diabetes mellitus (*n* = 175).

Variables	Low Muscle Mass		*p*	OR (95% CI)
	No (*n* = 129) *n* (%)	Yes (*n* = 46) *n* (%)		
Type 2 Diabetes Mellitus				
No (*n* = 101)	68 (67.3)	33 (32.7)	0.025	0.44 (0.21–0.91)
Yes (*n* = 74)	61 (82.4)	13 (17.6)		

Abbreviations: CI, confidence interval; OR, odds ratio.

**Table 4 medsci-13-00268-t004:** Risk of slow gait speed associated with physical inactivity (*n* = 175).

Variables	Slow Gait Speed		*p*	OR (95% CI)
	No (*n* = 157) *n* (%)	Yes (*n* = 18) *n* (%)		
Physical Activity Level				
Physically active (*n* = 87)	83 (95.4)	4 (4.6)	0.014	3.93 (1.24–12.45)
Insufficiently active (*n* = 88)	74 (84.1)	14 (15.9)		

Abbreviations: CI, confidence interval; OR, odds ratio.

**Table 5 medsci-13-00268-t005:** Risk of sarcopenia associated with type 2 diabetes mellitus (*n* = 175).

Variables	Sarcopenia Diagnosis		*p*	OR (95% CI)
	No (*n* = 135) *n* (%)	Yes (*n* = 40) *n* (%)		
Type 2 Diabetes Mellitus				
Yes (*n* = 101)	69 (68.3)	32 (31.7)	<0.001	0.26 (0.11–0.61)
No (*n* = 74)	66 (89.2)	8 (10.8)		

Abbreviations: CI, confidence interval; OR, odds ratio.

## Data Availability

The original contributions presented in this study are included in the article. Further inquiries can be directed to the corresponding author.

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
