# Peer review of "Sarcopenia and Functional Decline in Postmenopausal Women: The Roles of Type 2 Diabetes and Physical Activity"

_medsci, 2025, doi:10.3390/medsci13040268_

Round 1

Reviewer 1 Report

Comments and Suggestions for Authors

This study is a cross-sectional investigation aimed at exploring the independent and combined effects of type 2 diabetes mellitus (T2DM) and physical activity levels on sarcopenia and functional decline in postmenopausal women. The study enrolled 175 postmenopausal women, who were stratified into four groups based on T2DM status and physical activity level (≥150 minutes/week defined as active).  The study found that physical inactivity was the primary predictor of functional decline (gait speed <1.0 m/s); in unadjusted analyses, T2DM appeared to have a "protective effect" against sarcopenia, but after adjustment for obesity, both T2DM and obesity emerged as independent risk factors for sarcopenia. The authors propose that sarcopenia and functional decline may be driven by distinct pathological mechanisms. The research topic is clinically significant, and the findings offer some insightful implications for the comprehensive management of postmenopausal women with T2DM.

Main limitations:

  1. The diverse sample sources (social media, telephone invitations, existing cohort) may introduce selection bias.
  2. The specific parameters for the sample size calculation (e.g., effect size, expected proportions) were not provided, making it difficult to assess its
  3. The cross-sectional design cannot establish causality and can only suggest associations.
  4. The assessment of physical activity level partially relied on self-report, which is subject to recall bias.
  5. The types of exercise differed between the active groups (G1: resistance + aerobic; G3: primarily aerobic + water-based exercises), compromising comparability between groups.
  6. The paradoxical finding of T2DM appearing "protective" in the unadjusted model was not sufficiently explained.
  7. In the discussion, the explanation for the proposed "dichotomy" between "metabolic pathways" and "functional decline" lacks depth and is not well-supported by physiological or molecular-level evidence.

Author Response

RESPONSE TO REVIEWER 01,

GENERAL COMMENT: “This study is a cross-sectional investigation aimed at exploring the independent and combined effects of type 2 diabetes mellitus (T2DM) and physical activity levels on sarcopenia and functional decline in postmenopausal women. The study enrolled 175 postmenopausal women, who were stratified into four groups based on T2DM status and physical activity level (≥150 minutes/week defined as active).  The study found that physical inactivity was the primary predictor of functional decline (gait speed <1.0 m/s); in unadjusted analyses, T2DM appeared to have a "protective effect" against sarcopenia, but after adjustment for obesity, both T2DM and obesity emerged as independent risk factors for sarcopenia. The authors propose that sarcopenia and functional decline may be driven by distinct pathological mechanisms. The research topic is clinically significant, and the findings offer some insightful implications for the comprehensive management of postmenopausal women with T2DM.”

RESPONSE: We extend our sincere gratitude to Reviewer 1 for their comprehensive analysis and accurate summary of our study. We are pleased that the reviewer acknowledged the clinical significance of our research topic and recognized the value of our findings for the management of postmenopausal women with type 2 diabetes mellitus (T2DM).

COMMENT 1: “The diverse sample sources (social media, telephone invitations, existing cohort) may introduce selection bias.”

RESPONSE: We concur with the reviewer that this represents an important limitation. The heterogeneity of our recruitment sources introduces potential selection bias. We have now explicitly addressed this issue in the "Strengths, Limitations, and Future Directions" section (highlighted in yellow in Section 4.5).

COMMENT 2: “The specific parameters for the sample size calculation (e.g., effect size, expected proportions) were not provided, making it difficult to assess its.”

RESPONSE: We apologize for this critical omission. The reviewer is correct in noting that the input parameters for the sample size calculation were not provided. We have now revised Section 2.2 (Participants and Recruitment) to include the specific parameters entered into OpenEpi software. These parameters were derived from published literature on the expected prevalence differences for our primary outcomes.

COMMENT 3: “The cross-sectional design cannot establish causality and can only suggest associations.”

RESPONSE: This is a fundamental point, and we thank the reviewer for highlighting it. We fully agree that our use of causal language was inappropriate for a cross-sectional study design. To address this methodological flaw, we have taken two main actions:

Limitations Section: We have added this issue as the first limitation in our revised Section 4.5 ("Strengths, Limitations, and Future Directions"). The text now explicitly states: "First, the cross-sectional design precludes causal inference; we can only report associations rather than determine etiological pathways."

Systematic Manuscript Review: We conducted a comprehensive review of the entire manuscript (from the Abstract through the Conclusion) to replace all causal terminology (e.g., 'drive', 'determine', 'influence') with appropriate associative language (e.g., 'is associated with', 'predictor of', 'associational profiles').

The most substantial language modifications, beyond the Limitations Section, were made to the Abstract, the opening paragraph of the Discussion (Section 4), and the Conclusion (Section 5), which was completely rewritten to accurately reflect the associative nature of our data.

COMMENT 4: “The assessment of physical activity level partially relied on self-report, which is subject to recall bias.”

RESPONSE: This is a valid point. Self-reported physical activity represents a well-recognized limitation, as it is susceptible to recall bias and social desirability bias. We have expanded our discussion of this limitation in Section 4.5 to address this concern more explicitly and replaced our previous brief mention with a more comprehensive explanation.

COMMENT 5: “The types of exercise differed between the active groups (G1: resistance + aerobic; G3: primarily aerobic + water-based exercises), compromising comparability between groups.”

RESPONSE: We thank the reviewer for this crucial observation. While we acknowledge the heterogeneity in exercise modalities between active groups, we respectfully submit that this difference, rather than compromising comparability, provides a central insight of our study.

The data in Table 1 demonstrate that the G1 group (Active with T2DM), engaged in combined training (resistance + aerobic), exhibited superior appendicular skeletal muscle mass index (ASMI: 6.5 ± 0.8 kg/m²) compared to the G3 group (Active Normoglycemic) engaged in predominantly aerobic/aquatic exercise (ASMI: 5.7 ± 0.8 kg/m²), despite G1's T2DM status.

This finding is particularly noteworthy: individuals with T2DM who performed resistance training maintained higher muscle mass than metabolically healthy individuals who performed only aerobic exercise. This suggests that the anabolic stimulus from resistance training was sufficiently potent to counteract the catabolic effects of T2DM on skeletal muscle.

We have revised Section 4.1of the Discussion to explicitly highlight this finding. It strongly supports our conclusion that exercise modality (specifically the inclusion of resistance training) is critical for muscle mass preservation in this population, aligning with evidence from our cited references demonstrating the superiority of combined training approaches.

COMMENT 6: “The paradoxical finding of T2DM appearing "protective" in the unadjusted model was not sufficiently explained.”

RESPONSE: We thank the reviewer for highlighting this apparent contradiction. We agree that our original explanation for the seemingly “protective” association of T2DM was incomplete and warranted further clarification. To address this concern, we have substantially revised Section 4.1 of the manuscript (now entitled “The Sarcopenia Paradox: Associational Profiles of Obesity and Exercise Modality”). The revised section presents a more comprehensive, two-part explanation for this paradoxical finding:

  1. Statistical Confounding: We reiterate the role of sarcopenic obesity as a critical confounding variable that masks the true association between T2DM and sarcopenia in unadjusted models.
  2. Physiological Mechanism: We have expanded the discussion to incorporate the concept of ‘mass-function uncoupling’ (dynapenia). We argue that chronic hyperinsulinemia, characteristic of T2DM, exerts a potent anti-catabolic effect that may preserve muscle mass quantity. However, this preserved mass is qualitatively compromised—metabolically dysfunctional, insulin-resistant, and infiltrated with ectopic lipid deposits (myosteatosis). Thus, individuals with T2DM may maintain muscle mass but lose functional capacity, explaining the dissociation between sarcopenia prevalence and functional impairment observed in our data.

We believe this integrated physiological and statistical explanation, as detailed in the revised Section 4.1, now adequately addresses the complexity of this finding and resolves the apparent contradiction.

Comment 7: “In the discussion, the explanation for the proposed "dichotomy" between "metabolic pathways" and "functional decline" lacks depth and is not well-supported by physiological or molecular-level evidence.”

RESPONSE: This is an excellent point and highly constructive criticism. The reviewer is correct in noting that our original discussion of this dichotomy was conceptual rather than mechanistic.

To address this lack of mechanistic depth, we have substantially revised Section 4.3 (now titled “Integrating Metabolic and Neuromuscular Pathways”). The revised section now provides physiological and molecular-level support for these two distinct yet interactive pathways:

  1. The Metabolic Pathway (T2DM/Obesity -> Sarcopenia): We argue that this pathway is driven by systemic factors, including chronic low-grade inflammation (inflammaging), insulin resistance, and accumulation of Advanced Glycation End Products (AGEs). These factors lead to structural deterioration in both muscle quality (myosteatosis) and quantity (atrophy), ultimately compromising the tissue's metabolic and contractile properties.
  2. The Neuromuscular Pathway (Physical Inactivity -> Functional Decline): We argue that inactivity primarily impairs performance through neuromuscular deconditioning. This encompasses reduced motor unit firing rates, compromised neuromuscular junction integrity, and impaired intermuscular coordination. These neuromuscular deficits directly affect gait speed and functional performance, independent of preserved muscle mass.

We believe this revised section now provides the mechanistic depth the reviewer requested and clarifies how these pathways, while distinct in their primary mechanisms, may interact to produce the complex clinical phenotypes observed in our postmenopausal women.

We sincerely thank Reviewer 1 for the thorough evaluation of our manuscript and for recognizing its clinical significance and potential implications for the management of postmenopausal women with T2DM. We particularly appreciate the detailed and constructive feedback provided throughout the review.

The reviewer's astute observations on methodological rigor (specifically regarding the need to strictly adhere to associational language appropriate for our cross-sectional design, and the request for more comprehensive physiological explanations for the apparent T2DM paradox and the metabolic-functional dichotomy) were invaluable. These insights have enabled us to substantially strengthen the manuscript's clarity, methodological precision, and the mechanistic depth of our discussion.

We believe the revised manuscript now more accurately reflects the limitations inherent in cross-sectional research while providing a more robust physiological framework for interpreting our findings. We are confident these revisions have significantly enhanced the manuscript's scientific rigor and clinical applicability.

Reviewer 2 Report

Comments and Suggestions for Authors

I have reviewed the manuscript titled “Sarcopenia and Functional Decline in Postmenopausal Women: The Roles of Type 2 Diabetes and Physical Activity”. Overall, the study addresses an important topic, and it is well written. However, the following revisions are recommended to improve the quality of the manuscript:

- Please check if the abstract is within the limit of 200-250 words. Otherwise, reduce it.

- Line 73: the extended “T2DM” has already been cited previously.

- In my opinion, the Introduction section could be expanded, and the aims should be described more deeply.

- I would suggest adding a schematic flow-chart of the study, preferably following the CONSORT PRO scheme.

- The classification of physical activity is based on self-report. This limitation should be discussed more explicitly.

- Why do you use the Asian Working Group for Sarcopenia (AWGS 2019) criteria in a Brazilian population? Please discuss it.

- I think there is a mistake in some SD values of Tables 1 (11.2 ± 49 should be 4.9?)

- The References’ style doesn’t seem to follow the MDPI’s guidelines. Please revise it.

Author Response

RESPONSE TO REVIEWER 02,

GENERAL COMMENT: I have reviewed the manuscript titled “Sarcopenia and Functional Decline in Postmenopausal Women: The Roles of Type 2 Diabetes and Physical Activity”. Overall, the study addresses an important topic, and it is well written. However, the following revisions are recommended to improve the quality of the manuscript:”

RESPONSE: We thank Reviewer 2 for the careful evaluation of our manuscript. We are pleased that the reviewer acknowledged that our study addresses an important topic and found the manuscript to be well written. We agree that the manuscript can be further strengthened and are prepared to address each recommended revision point by point to enhance the quality and rigor of our work. Below, we provide our detailed responses to each comment, with corresponding revisions highlighted in the manuscript.

COMMENT 1: “Please check if the abstract is within the limit of 200-250 words. Otherwise, reduce it.”

RESPONSE: We thank Reviewer 2 for the attention to the journal's formatting guidelines. We have verified the word count of our revised abstract (the same version updated in response to Reviewers 1 and 3 to employ strictly associative language appropriate for our cross-sectional design). The revised abstract contains 242 words, which is within the journal's 250-word limit specified in the Instructions for Authors.

COMMENT 2: “Line 73: the extended “T2DM” has already been cited previously.”

RESPONSE: We thank Reviewer 2 for identifying this redundancy. We have corrected the duplicate definition of T2DM on line 73 as requested. Furthermore, we conducted a comprehensive review of the entire manuscript to identify similar issues and corrected an additional redundant acronym definition on line 80. We have now ensured that all abbreviations are defined only at their first occurrence throughout the manuscript, in accordance with standard scientific writing conventions.

COMMENT 3: “In my opinion, the Introduction section could be expanded, and the aims should be described more deeply.”

RESPONSE: We agree with Reviewer 2's assessment. Our original introduction was too brief, and the study objectives were not described with sufficient depth and clarity.

To address this concern, we have substantially revised Section 1 (Introduction). We expanded the contextual background to include data on the global prevalence of T2DM and its specific impact on exacerbating sarcopenia, drawing from contextual information in the original master's dissertation and references already cited in our Discussion section.

Most importantly, we completely rewrote the final paragraphs of the Introduction. These revised paragraphs now provide a more comprehensive description of the knowledge gap, specifically the uncertainty regarding the distinct roles of type 2 diabetes mellitus (T2DM) versus physical inactivity in driving sarcopenia and functional decline. They also explicitly state the study's specific objectives, including the analysis of independent predictors and the investigation of obesity as a potential confounding variable. We believe the revised Introduction now provides a much stronger foundation for readers to understand and contextualize the results and discussion that follow.

COMMENT 4: “I would suggest adding a schematic flow-chart of the study, preferably following the CONSORT PRO scheme.”

RESPONSE: We sincerely appreciate Reviewer 2's insightful suggestion to improve methodological transparency through a flow diagram. We recognize that the reviewer referenced the CONSORT PRO scheme, which indeed represents the benchmark for reporting patient-reported outcomes in randomized controlled trials. However, given that our investigation employs a cross-sectional observational design rather than an interventional approach, we have developed a participant flow diagram adhering to the STROBE (Strengthening the Reporting of Observational Studies in Epidemiology) guidelines—the internationally recognized reporting standard for observational research. This framework, previously cited in Section 2.1 (Study Design), provides specific recommendations for transparently reporting participant recruitment, classification, and analysis in cross-sectional studies. The flow diagram has been incorporated into the revised manuscript to enhance clarity regarding our participant selection procedures.

COMMENT 5: “The classification of physical activity is based on self-report. This limitation should be discussed more explicitly.”

RESPONSE: We appreciate Reviewer 2's identification of this important methodological limitation and fully acknowledge its significance. Notably, this concern was raised independently by multiple reviewers, underscoring its relevance to our study design. In response, we have substantially revised and expanded Section 4.5 (Strengths, Limitations, and Future Directions) to provide a more comprehensive discussion of measurement-related limitations. The revised manuscript now explicitly addresses the potential for recall bias and social desirability bias inherent in self-reported physical activity assessment, which may have led participants to overestimate their activity levels or provide socially desirable responses. We further acknowledge that objective measurement methods, particularly accelerometry-based approaches, would minimize these biases and provide more accurate quantification of physical activity patterns. This has been highlighted as a priority for future research in this population.

COMMENT 6: “Why do you use the Asian Working Group for Sarcopenia (AWGS 2019) criteria in a Brazilian population? Please discuss it.”

RESPONSE: We sincerely appreciate Reviewer 2's critical observation regarding the application of AWGS 2019 criteria to a Brazilian population. This was indeed a deliberate and carefully considered methodological decision that merits explicit justification. In response to this important concern, we have added a comprehensive explanatory paragraph in Section 2.5.3 (Sarcopenia Diagnosis) that addresses this rationale. Our selection of AWGS 2019 criteria was based on two interconnected methodological considerations:

  1. Absence of Validated National Criteria: Currently, Brazil lacks a nationally standardized and universally adopted diagnostic consensus for sarcopenia assessment, particularly in postmenopausal women. This limitation requires researchers to judiciously select from established international guidelines. While multiple international consensus statements exist (EWGSOP2, AWGS 2019, IWGS), the methodological challenge lies in selecting the framework most appropriate for the specific research question and target population.
  2. Enhanced Sensitivity Through Conservative Functional Thresholds: Although EWGSOP2 is frequently applied in Brazilian studies, we selected AWGS 2019 specifically because its functional cutoff points demonstrate greater sensitivity for detecting early-stage functional decline. The AWGS 2019 criteria employ higher thresholds for both handgrip strength (<18 kg vs. <16 kg in EWGSOP2) and gait speed (<1.0 m/s vs. ≤0.8 m/s in EWGSOP2).

Given that our primary research objective centers on identifying early functional impairment in a high-risk population of postmenopausal women (particularly those with metabolic dysfunction associated with type 2 diabetes) this more conservative cutoff points align better with our study's focus on detecting incipient rather than advanced sarcopenia. The higher thresholds potentially capture individuals at earlier stages of functional decline, when preventive interventions may be most effective.

COMMENT 7: “I think there is a mistake in some SD values of Tables 1 (11.2 ± 49 should be 4.9?)”

RESPONSE: We sincerely thank Reviewer 2 for their exceptional attention to detail in identifying the typographical error in Table 1. The reviewer was absolutely correct; the standard deviation value of '49' for Fat Mass Index in Group 3 (G3) was indeed a transcription error.

The reviewer's keen observation prompted us to conduct a comprehensive audit and verification of all data presented in Table 1 against our original raw dataset. During this thorough review process, we regrettably identified that, in addition to the error initially flagged, several other transcription errors had gone undetected—specifically in the mean and/or standard deviation values for Age, Fat Mass Index, Fat Mass %, Handgrip Strength, and Gait Speed in Group 4 (G4).

We are deeply grateful for this reviewer's intervention, which proved crucial in ensuring the accuracy and integrity of our data presentation. The revised Table 1 in the manuscript now accurately reflects the data from our sample. All corrections have been highlighted in the revised manuscript for transparency.

COMMENT 8: “The References’ style doesn’t seem to follow the MDPI’s guidelines. Please revise it.”

RESPONSE: We thank Reviewer 2 for this observation and apologize for any formatting inconsistencies in the reference list. Our bibliography was initially generated using Mendeley Desktop reference management software with the MDPI style file.

In response to the reviewer's request, we have conducted a meticulous manual review of the entire reference list in this revised version. All entries have been carefully verified and corrected to ensure strict adherence to the specific style guidelines of Medical Sciences. We appreciate the reviewer's attention to detail, which has helped us improve the overall quality and consistency of our manuscript.

We sincerely thank Reviewer 2 for their positive and constructive feedback. We were particularly encouraged by the reviewer's assessment that our study is 'well written' and 'addresses an important topic'.

The reviewer's recommendations were instrumental in improving the manuscript's quality. The suggestions to expand the Introduction and add a participant flowchart have significantly enhanced the paper's clarity and logical structure.

We are especially grateful for the reviewer's meticulous attention to detail, particularly in identifying the typographical error in Table 1. This critical observation not only enabled correction of the initially identified error but also prompted us to conduct a comprehensive data audit, which revealed and allowed us to correct several additional transcription inaccuracies.

Furthermore, the request for explicit methodological justification regarding the use of AWGS 2019 criteria in our Brazilian population has substantially strengthened the scientific rigor of our Methods section. We are grateful for all these valuable contributions, which have markedly improved the overall quality of our manuscript.

Reviewer 3 Report

Comments and Suggestions for Authors

The article “ Sarcopenia and Functional Decline in Postmenopausal Women: 2 The Roles of Type 2 Diabetes and Physical Activity 3 ” is a clear investigation of independent and combined effects of Type 2 Diabetes Mellitus (T2DM) and physical activity on sarcopenia and functional performance in postmenopausal women in a Cross-sectional Study.

This study employs a cross-sectional design, assessing 175 women stratified by diabetes status and activity level. Body composition was measured via DXA, muscle strength by handgrip dynamometry, and gait speed as a marker of functional performance. The methodology aligns with standard approaches in clinical and geriatric physiology research.

The use of the Asian Working Group for Sarcopenia 2019 criteria is commendable, ensuring diagnostic consistency. However, a cross-sectional approach limits causal inference.

Although the authors state that sample size was calculated a priori using OpenEpi, the statistical power of 80% might not fully capture complex interactions among T2DM, obesity, and physical activity. Therefore, while the study is methodologically appropriate, it is constrained by inherent design limitations.

This study is relevant and timely, focusing on sarcopenia in postmenopausal women — a demographic with heightened metabolic and musculoskeletal vulnerability. While sarcopenia and diabetes interactions have been explored previously, this article emphasizes the interplay between T2DM, obesity, and activity, elucidating their distinct roles in muscle and functional decline. The study’s originality lies in disentangling the metabolic versus behavioral determinants of sarcopenia, thus addressing a genuine research gap.

This article contributes by highlighting the differential pathways through which T2DM and inactivity affect muscle health and functionality, but this is a cross-sectional study. It demonstrates that T2DM and obesity are metabolic determinants of sarcopenia, while inactivity primarily predicts functional impairment. This conceptual distinction enriches current understanding and supports precision in exercise prescription for diabetic postmenopausal women. Nonetheless, its contribution remains incremental rather than transformative, reinforcing rather than redefining the existing paradigm. In the same time, this is a transversal study too. This is not an observational or interventional study, all observations could be only regarding the association, not determination.

Future studies should integrate objective activity monitoring (e.g., accelerometry) instead of self-reported activity levels to minimize recall bias.

Additional confounders — such as diet, vitamin D levels, inflammatory biomarkers, and hormonal profiles — should be included, given their established influence on muscle metabolism, but only the association could be observed, as I already said.

The conclusions are not internally consistent and logically aligned with the results. The association not determination could be evaluated.

The authors correctly note the distinct etiologies of sarcopenia and functional decline. However, statements implying causation exceed what can be justified by cross-sectional data.

While the dual intervention approach — metabolic optimization and structured exercise — is plausible, the sustainability of these recommendations is not empirically tested in this study. Hence, the conclusions, though coherent, remain hypothetical pending a new and other longitudinal validated study.

The reference list is extensive and current, including seminal works and recent meta-analyses (2022–2025). The inclusion of consensus documents such as the AWGS 2019 criteria and ESPEN/EASO statements enhances credibility. However, there is mild overreliance on national Brazilian sources, which could limit the paper’s international generalizability.

Tables are generally clear and well-structured.

Table 1 effectively summarizes baseline differences, though overlapping standard deviations suggest group homogeneity.

Tables 3–5 correctly present odds ratios, yet confidence intervals are wide, indicating limited precision. Including visual figures (e.g., forest plots or scatter diagrams) could have improved interpretability and reader engagement.

The manuscript is simply written, logically organized, and adheres to STROBE guidelines – but only association could be expressed. Not determinant, not causes, not other aspects that could be judged as determinants.

The authors express an acknowledgment of AI-assisted editing work. This fact is transparent.

Ethically, the study is sound. However, inclusion bias may have occurred since active participants were recruited from a university-affiliated program, potentially representing a healthier subgroup. In the same time the same aspect – only association could be presented.

This article offers a solid transversal observational contribution to understanding sarcopenia in postmenopausal women with T2DM, emphasizing behavioral and metabolic distinctions, but for only for this group, only for these aspects.

Its methodological transparency, ethical rigor, and comprehensive discussion make it suitable for publication after revisions, especially in adapt methodological transversal study technique and refining interpretive claims.

Future studies should integrate objective activity monitoring (e.g., accelerometry) instead of self-reported activity levels to minimize recall bias.

Additional confounders — such as diet, vitamin D levels, inflammatory biomarkers, and hormonal profiles — should be included, given their established influence on muscle metabolism, but only the association could be observed, as I already said.

The conclusions are not internally consistent and logically aligned with the results. The association not determination could be evaluated.

The authors correctly note the distinct etiologies of sarcopenia and functional decline. However, statements implying causation exceed what can be justified by cross-sectional data.

While the dual intervention approach — metabolic optimization and structured exercise — is plausible, the sustainability of these recommendations is not empirically tested in this study. Hence, the conclusions, though coherent, remain hypothetical pending a new and other longitudinal validated study.

The reference list is extensive and current, including seminal works and recent meta-analyses (2022–2025). The inclusion of consensus documents such as the AWGS 2019 criteria and ESPEN/EASO statements enhances credibility. However, there is mild overreliance on national Brazilian sources, which could limit the paper’s international generalizability.

Tables are generally clear and well-structured.

Table 1 effectively summarizes baseline differences, though overlapping standard deviations suggest group homogeneity.

Tables 3–5 correctly present odds ratios, yet confidence intervals are wide, indicating limited precision. Including visual figures (e.g., forest plots or scatter diagrams) could have improved interpretability and reader engagement.

The manuscript is simply written, logically organized, and adheres to STROBE guidelines – but only association could be expressed. Not determinant, not causes, not other aspects that could be judged as determinants.

The authors express an acknowledgment of AI-assisted editing work. This fact is transparent.

Ethically, the study is sound. However, inclusion bias may have occurred since active participants were recruited from a university-affiliated program, potentially representing a healthier subgroup. In the same time the same aspect – only association could be presented.

This article offers a solid transversal observational contribution to understanding sarcopenia in postmenopausal women with T2DM, emphasizing behavioral and metabolic distinctions, but for only for this group, only for these aspects.

Its methodological transparency, ethical rigor, and comprehensive discussion make it suitable for publication after revisions, especially in adapt methodological transversal study technique and refining interpretive claims.

Author Response

RESPONSE TO REVIEWER 03,

GENERAL COMMENT: The article “Sarcopenia and Functional Decline in Postmenopausal Women: The Roles of Type 2 Diabetes and Physical Activity” is a clear investigation of independent and combined effects of Type 2 Diabetes Mellitus (T2DM) and physical activity on sarcopenia and functional performance in postmenopausal women in a Cross-sectional Study.

This study employs a cross-sectional design, assessing 175 women stratified by diabetes status and activity level. Body composition was measured via DXA, muscle strength by handgrip dynamometry, and gait speed as a marker of functional performance. The methodology aligns with standard approaches in clinical and geriatric physiology research.

RESPONSE: We thank Reviewer 3 for an exceptionally thorough, insightful, and methodologically rigorous review. The reviewer's comments have been instrumental in improving our manuscript, and their emphasis on aligning our interpretative language with the cross-sectional nature of our study design served as the catalyst for the most significant and important revisions to the manuscript.

COMMENT 1: The use of the Asian Working Group for Sarcopenia 2019 criteria is commendable, ensuring diagnostic consistency. However, a cross-sectional approach limits causal inference.

RESPONSE: We thank Reviewer 3 for this accurate summary and for their positive comments. We were particularly encouraged to see that the reviewer considered our methodology aligned with 'standards in clinical physiology and geriatric research' and deemed our use of the AWGS 2019 criteria 'commendable'.

We acknowledge that the “a cross-sectional approach limits causal inference,” which represents the most critical and fundamental methodological critique of the review. We fully agree with Reviewer 3 that our use of causal language (e.g., 'drive', 'determinant', 'etiological pathways') was methodologically inappropriate and indefensible for a cross-sectional study design. To address this fundamental flaw, we have taken the following comprehensive actions:

  1. Systematic Language Revision: We conducted a thorough review of the entire manuscript (from Abstract to Conclusion) replacing all causal terminology with appropriate associative language (e.g., 'is associated with', 'correlates with', 'associational profiles').
  2. Complete Rewriting of Conclusions: Section 5 (Conclusions) has been entirely rewritten. As the reviewer requested, the new conclusion is now 'logically aligned with the findings', focusing strictly on the associations observed and the hypothetical clinical implications derived from them, rather than asserting causal 'etiologies'.
  3. Explicit Statement of Limitation: We have added this issue as the first and primary limitation in the revised Section 4.5 (Limitations), which now explicitly states: 'First, the cross-sectional design precludes causal inference. We can only report robust associations, not determine etiological pathways'.

We believe the manuscript now rigorously adheres to the standards appropriate for an 'observational cross-sectional study' and accurately reflects the limitations inherent in this study design.

COMMENT 2: Although the authors state that sample size was calculated a priori using OpenEpi, the statistical power of 80% might not fully capture complex interactions among T2DM, obesity, and physical activity. Therefore, while the study is methodologically appropriate, it is constrained by inherent design limitations.

RESPONSE: This is a valid observation. In response to this concern (which was also raised by Reviewer 1), we have:

  1. Revised Section 2.2 to include the specific a priori input parameters (expected prevalences of 30% vs. 10%) that informed our sample size calculation.
  2. Added a new limitation in Section 4.5 that explicitly acknowledges this point. The text now states that the 'wide confidence intervals' (as the reviewer also noted) indicate 'limited precision', and while statistical power was adequate for the primary outcomes, the sample may be underpowered to capture complex interactions or detect smaller effect sizes.

COMMENT 3: This study is relevant and timely, focusing on sarcopenia in postmenopausal women — a demographic with heightened metabolic and musculoskeletal vulnerability. While sarcopenia and diabetes interactions have been explored previously, this article emphasizes the interplay between T2DM, obesity, and activity, elucidating their distinct roles in muscle and functional decline. The study’s originality lies in disentangling the metabolic versus behavioral determinants of sarcopenia, thus addressing a genuine research gap.

This article contributes by highlighting the differential pathways through which T2DM and inactivity affect muscle health and functionality, but this is a cross-sectional study. It demonstrates that T2DM and obesity are metabolic determinants of sarcopenia, while inactivity primarily predicts functional impairment. This conceptual distinction enriches current understanding and supports precision in exercise prescription for diabetic postmenopausal women. Nonetheless, its contribution remains incremental rather than transformative, reinforcing rather than redefining the existing paradigm. In the same time, this is a transversal study too. This is not an observational or interventional study, all observations could be only regarding the association, not determination.

Future studies should integrate objective activity monitoring (e.g., accelerometry) instead of self-reported activity levels to minimize recall bias.

RESPONSE: We thank Reviewer 3 for this insightful analysis and for recognizing the relevance and originality of our study. We are very pleased that the reviewer identified the strength of our work as 'disentangling metabolic versus behavioral determinants' and that this 'conceptual distinction enriches current understanding.' The reviewer is absolutely correct in identifying the two primary methodological concerns:

  1. Causal Language (Association vs. Determination): This was the most critical observation. We fully agree that our original use of causal language (e.g., 'determinants,' 'drives') was inappropriate for a cross-sectional study. In response, we conducted a systematic review of the entire manuscript—from Abstract through Discussion to Conclusion—replacing all causal terminology with appropriate associative language (e.g., 'associated with,' 'predictor of,' 'associational profiles'). We have also added this issue as the first and primary limitation in Section 4.5 (Limitations), explicitly stating that cross-sectional designs preclude causal inference. We believe the article now accurately reflects the observational nature of our data, as the reviewer requested.
  2. Self-Report Bias (Physical Activity Assessment): We also fully agree regarding the need for objective monitoring. As the reviewer notes, self-report is susceptible to recall and social desirability bias. We have addressed this directly in Section 4.5 (Limitations), where we now explicitly discuss recall bias and the need for future research integrating objective monitoring (e.g., accelerometry) to minimize this bias.

COMMENT 4: Additional confounders — such as diet, vitamin D levels, inflammatory biomarkers, and hormonal profiles — should be included, given their established influence on muscle metabolism, but only the association could be observed, as I already said.

RESPONSE: We agree. This is an important omission. In response, we have expanded Section 4.5 (Limitations) to explicitly state that we did not assess 'dietary factors (especially protein intake), vitamin D status, or inflammatory markers' and acknowledge that these represent important potential confounding variables that should be incorporated in future research on this topic.

COMMENT 5: The conclusions are not internally consistent and logically aligned with the results. The association not determination could be evaluated.

RESPONSE: We thank Reviewer 3 for this fundamental critique. We fully agree. You is correct in noting that our original Conclusion (Section 5) was not 'internally consistent and logically aligned with the results.' This inconsistency was a direct result of our inappropriate use of causal language (e.g., 'determinants') to describe data from a cross-sectional study, where only 'association' can be assessed.

To correct this central flaw, we have completely rewritten Section 5 (Conclusions). The new conclusion has been meticulously revised to remove all causal terminology and replace it with strictly associative language (e.g., 'associational profiles,' 'associated with,' 'predictor of'). We believe the revised Conclusion is now fully aligned with our results and accurately reflects the observational nature of our data, as requested by the reviewer.

COMMENT 6: The authors correctly note the distinct etiologies of sarcopenia and functional decline. However, statements implying causation exceed what can be justified by cross-sectional data.

While the dual intervention approach — metabolic optimization and structured exercise — is plausible, the sustainability of these recommendations is not empirically tested in this study.

RESPONSE: We thank Reviewer 3 for these two interconnected observations, which are central to improving our manuscript. The reviewer is absolutely correct on both points.

  1. Regarding Causal Language: We fully agree that our use of terms such as 'etiologies' and other language implying causation exceeded what can be justified by cross-sectional data. Cross-sectional studies, by their nature, measure exposure and outcome simultaneously and therefore cannot establish temporal precedence or causal relationships. To correct this fundamental flaw, we conducted a systematic review of the entire manuscript—from Abstract through Discussion—replacing causal terminology with strictly associative language (e.g., 'associational profiles,' 'predictor of,' 'associated with').
  2. Regarding Hypothetical Conclusions: As a direct consequence of the correction above, we also fully agree that our intervention recommendations, while 'plausible,' are 'hypothetical' and untested. Our original Section 5 (Conclusions) failed to make this clear. Therefore, we have completely rewritten Section 5. The new conclusion now explicitly frames our 'dual intervention' approach as a logical implication of our associational findings, not as an empirically tested conclusion. The revised text now concludes by stating that 'although this cross-sectional design precludes causal inference, our findings provide a strong rationale for longitudinal studies,' aligning perfectly with the reviewer's critique.

COMMENT 7: Hence, the conclusions, though coherent, remain hypothetical pending a new and other longitudinal validated study.

RESPONSE: We thank Reviewer 3 for this observation, with which we fully agree.

The reviewer is absolutely correct. Our conclusions regarding the 'dual-target approach' (metabolic and functional) are indeed 'hypothetical' and represent a logical implication of our associational findings, not an empirically tested recommendation.

In direct response to this fundamental critique (which is linked to previous comments regarding causality), we have completely rewritten Section 5 (Conclusions). The new conclusion now explicitly ends with a statement that precisely reflects the reviewer's caveat, stating: 'Although this cross-sectional design precludes causal inference, our findings provide a strong rationale for longitudinal studies and clinical trials...', thereby acknowledging the need for 'a validated longitudinal study' to confirm our hypotheses.

COMMENT 8: The reference list is extensive and current, including seminal works and recent meta-analyses (2022–2025). The inclusion of consensus documents such as the AWGS 2019 criteria and ESPEN/EASO statements enhances credibility. However, there is mild overreliance on national Brazilian sources, which could limit the paper’s international generalizability.

RESPONSE: We appreciate this observation. In the revision, we have made a deliberate effort to ensure that our core mechanistic interpretations (e.g., in Sections 4.1 and 4.3) are supported by high-impact international references. We have retained Brazilian sources only when essential for describing national health guidelines or the local exercise program ('Doce Vida'), which we believe is methodologically appropriate given the study context and relevant for readers interested in the specific intervention setting.

COMMENT 9: Tables are generally clear and well-structured.

RESPONSE: We thank Reviewer 3 for this comment.

COMMENT 10: Table 1 effectively summarizes baseline differences, though overlapping standard deviations suggest group homogeneity.

RESPONSE: We thank Reviewer 3 for this precise statistical observation. You are entirely correct. As demonstrated by the non-significant p-values for variables such as Age (p=0.200), Body Mass Index (p=0.170), Fat Mass Index (p=0.670), and Fat Mass Percentage (p=0.200), the overlapping standard deviations indicate that our four groups were indeed highly homogeneous in their demographic and overall adiposity characteristics.

​We consider this baseline homogeneity a methodological strength rather than a limitation. It suggests that the groups were well-matched with respect to important potential confounding factors (age and adiposity), which could otherwise have obscured our findings.

​This homogeneity actually strengthens our primary findings. The fact that highly significant differences emerged in our primary outcomes—specifically in Appendicular Skeletal Muscle Mass Index (ASMI, p=0.002), Handgrip Strength (p=0.008), and Gait Speed (p=0.002) – despite group similarity in age and adiposity, reinforces our conclusion that these differences are robustly associated with our stratification criteria (T2DM status and physical activity level) rather than being artifacts of demographic or adiposity imbalances between groups.

COMMENT 11: Tables 3–5 correctly present odds ratios, yet confidence intervals are wide, indicating limited precision.

RESPONSE: This is an insightful and entirely correct statistical observation. In our revised Section 4.5 (Limitations), we have added a new statement (now Point 4) that directly addresses this issue. The text explicitly acknowledges that the 'wide 95% confidence intervals' observed in some analyses indicate 'limited precision' and states that 'the true magnitude of the effect is uncertain and should be interpreted with caution.'

COMMENT 12: Including visual figures (e.g., forest plots or scatter diagrams) could have improved interpretability and reader engagement.

RESPONSE: We appreciate the reviewer's suggestion and carefully considered adding graphical representations, such as forest plots, for the Odds Ratios presented in Tables 3-5.

However, as the reviewer also astutely noted, our confidence intervals for these ORs are indeed 'wide.' We believe that presenting these ORs with such wide CIs in a graphical format (such as a forest plot) could paradoxically suggest visually a level of precision that the data do not support. Forest plots are most informative when confidence intervals are relatively narrow, as they allow clear visual interpretation of effect magnitude and statistical significance.

Given that the exact numerical data are already clearly presented in Tables 3, 4, and 5, and that we have already added the new participant flowchart (Figure 1) as requested by Reviewer 2, we prefer (for reasons of methodological rigor and transparency) to maintain the presentation of regression results in tabular format in this version. We believe this approach more honestly represents the uncertainty in our estimates.

COMMENT 13: The manuscript is simply written, logically organized, and adheres to STROBE guidelines – but only association could be expressed. Not determinant, not causes, not other aspects that could be judged as determinants.

The authors express an acknowledgment of AI-assisted editing work. This fact is transparent.

RESPONSE: We thank Reviewer 3 for this observation. We are very pleased that the reviewer found the manuscript 'plainly written, logically organized, and adherent to STROBE guidelines'. We also appreciate the recognition of our 'transparency' regarding the 'acknowledgement of AI-assisted editing work.'

The central and most critical point of this comment—that 'only association could be expressed. Not determinants, not causes...' – was, in our assessment, the most important and constructive critique of the entire review process.

We agree completely. As this represented the central methodological flaw, we conducted a systematic review of the entire manuscript (from Abstract through Discussion, including a complete rewrite of the Conclusion) to correct precisely this issue.

All language implying causality (such as 'determinant,' 'causes,' and 'drives') was meticulously removed and replaced with appropriate associative terminology (such as 'predictor of,' 'associated with,' 'correlates with'). We believe the revised manuscript now strictly adheres to the interpretative limitations of a cross-sectional study design, as the reviewer appropriately required.

COMMENT 14: Ethically, the study is sound. However, inclusion bias may have occurred since active participants were recruited from a university-affiliated program, potentially representing a healthier subgroup. In the same time the same aspect – only association could be presented.

RESPONSE: We fully agree. This is the same selection bias concern raised by Reviewer 1. As previously addressed in our response to that reviewer, we have added this issue as an explicit limitation in Section 4.5, acknowledging the 'healthy volunteer bias' inherent in recruitment from the 'Doce Vida' program and stating that this 'limits the generalizability of our findings' to the broader postmenopausal population.

COMMENT 15: This article offers a solid transversal observational contribution to understanding sarcopenia in postmenopausal women with T2DM, emphasizing behavioral and metabolic distinctions, but for only for this group, only for these aspects.

Its methodological transparency, ethical rigor, and comprehensive discussion make it suitable for publication after revisions, especially in adapt methodological transversal study technique and refining interpretive claims.

RESPONSE: We fully agree. This is the same selection bias concern raised by Reviewer 1. As previously addressed in our response to that reviewer, we have added this issue as an explicit limitation in Section 4.5, acknowledging the 'healthy volunteer bias' inherent in recruitment from the 'Doce Vida' program and stating that this 'limits the generalizability of our findings' to the broader postmenopausal population.

We thank Reviewer 3 once again for their incisive and methodologically rigorous review. The reviewer's insistence on refining our interpretative language catalyzed the most significant improvement to our manuscript, and we believe the article is substantially stronger and more defensible because of their critical feedback.

Round 2

Reviewer 1 Report

Comments and Suggestions for Authors

The main concerns have been addressed.

Reviewer 3 Report

Comments and Suggestions for Authors

My insistence on refining your interpretative language served as the catalyst for the
most significant improvement of manuscript. The article is substantially
stronger and more defensible because of their critical feedback.

Most of suggestions are implemented. The article could be published, as it is. 

Best